# Effect of Weight Loss after Bariatric Surgery on Thyroid-Stimulating Hormone Levels in Euthyroid Patients with Morbid Obesity

**DOI:** 10.3390/nu11051121

**Published:** 2019-05-20

**Authors:** Paula Juiz-Valiña, Elena Outeiriño-Blanco, Sonia Pértega, Bárbara María Varela-Rodríguez, María Jesús García-Brao, Enrique Mena, Lara Pena-Bello, María Cordido, Susana Sangiao-Alvarellos, Fernando Cordido

**Affiliations:** 1Endocrine, Nutritional and Metabolic Diseases Group, Faculty of Health Sciences, University of A Coruña, 15006 A Coruña, Spain; Paula.Juiz.Valina@sergas.es (P.J.-V.); barbara.varela.rodriguez@gmail.com (B.M.V.-R.); Maria.Lara.Pena.Bello@sergas.es (L.P.-B.); Maria.Cordido.Carro@sergas.es (M.C.); 2Instituto de Investigación Biomedica (INIBIC), University Hospital A Coruña, 15006 A Coruña, Spain; 3Department of Endocrinology, University Hospital A Coruña, 15006 A Coruña, Spain; Elena.Outeirino.Blanco@sergas.es; 4Clinical Epidemiology and Biostatistics Unit, University Hospital A Coruña, 15006 A Coruña, Spain; Sonia.Pertega.Diaz@sergas.es; 5Department of Digestive and General Surgery, University Hospital A Coruña, 15006 A Coruña, Spain; MA.Jesus.Garcia.Brao@sergas.es (M.J.G.-B.); Enrique.Mena.del.Rio@sergas.es (E.M.)

**Keywords:** obesity, weight loss, bariatric surgery, thyroid, TSH

## Abstract

Obesity is associated with several endocrine abnormalities, including thyroid dysfunction. The objective of this study was to investigate the effect of weight loss after bariatric surgery on thyroid-stimulating hormone (TSH) levels in euthyroid patients with morbid obesity. We performed an observational study, evaluating patients with morbid obesity submitted to bariatric surgery. We included 129 patients (92 women) and 31 controls (21 women). Clinical, anthropometric, biochemical, and hormonal parameters were evaluated. The primary endpoint was circulating TSH (µU/mL). Fasting TSH levels were higher in the obese group (3.3 ± 0.2) than in the control group (2.1 ± 0.2). The mean excessive body mass index (BMI) loss (EBMIL) 12 months after bariatric surgery was 72.7 ± 2.1%. TSH levels significantly decreased in the obese patients after surgery; 3.3 ± 0.2 vs. 2.1 ± 0.2 before and 12 months after surgery, respectively. Free thyroxine (T4) (ng/dL) levels significantly decreased in the obese patients after surgery; 1.47 ± 0.02 vs. 1.12 ± 0.02 before and 12 months after surgery, respectively. TSH decreased significantly over time, and the decrement was associated with the EBMIL. In euthyroid patients with morbid obesity, weight loss induced by bariatric surgery promotes a significant decline of the increased TSH levels. This decrement of TSH is progressive over time after surgery and significantly associated with excess BMI loss.

## 1. Introduction

Obesity is a major public health problem. Since 1980, the prevalence of obesity has doubled in more than 70 countries and has continuously increased in most other countries. A high body mass index (BMI) accounts for some 4.0 million deaths globally. More than two-thirds of deaths related to high BMI were due to cardiovascular disease. The disease burden related to high BMI has increased since 1990 [1]. The age-adjusted prevalence of obesity in the United States in 2013–2014 was 35.0% among men and 40.4% among women. The corresponding values for class three obesity were 5.5% for men and 9.9% for women [2]. Similar, although slightly lower, results are found in Europe [3] and worldwide [4]. Moderate 5% weight loss improves metabolic function in multiple organs simultaneously, and progressive weight loss causes dose-dependent alterations in key adipose tissue biological pathways [5]. Among obese patients, bariatric surgery (BS) using laparoscopic banding, gastric bypass, or laparoscopic sleeve gastrectomy, compared with usual care nonsurgical obesity management, was associated with more marked improvement in several comorbidities and lower all-cause mortality [6].

Adiposity is associated with several endocrine abnormalities, including decreased stimulated growth hormone (GH) secretion [7,8,9] and thyroid dysfunction [9,10]. Thyroid hormone levels have been reported to be normal, increased, and decreased in obese patients; this discrepancy probably reflects the fact that patients were examined at different times and may differ in degree and type of obesity and plasma insulin resistance [10,11,12,13,14,15]. The alteration of thyroid function and the effect of BS on postoperative thyroid function evolution are still not completely understood. There are several previous studies showing different results regarding the variation of thyroid-stimulating hormone (TSH) after BS and the relation of TSH variation with weight loss [15,16,17,18,19,20]. In addition, the mechanism and the clinical implications of this hormonal alteration remains unknown. 

The objective of this study was to investigate the effect of weight loss after bariatric surgery on TSH circulating levels in morbid obese euthyroid patients.

## 2. Patients and Methods

### 2.1. Patients and Controls

This study was conducted in accordance with the Declaration of Helsinki. The study protocol was approved by the research ethics committee of Galicia, Spain; written informed consent was obtained from all patients and controls. We included a total of 160 patients and controls (113 women) in our study, including 129 patients (92 women) and 31 controls (21 women) selected from a pool of volunteers available to our unit. Prior to surgery 51% of the obese patients had diabetes. None of the controls had diabetes mellitus or other medical problems, nor were they taking any drugs.

We accomplished a retrospective observational study evaluating patients with morbid obesity and normal preoperative thyroid function submitted to bariatric surgery in our hospital between January 2016 and December 2018. Patients were excluded if they had a history of thyroid disease, treatment with thyroid hormone, or treatment with antithyroid drugs, amiodarone or lithium. 

### 2.2. Study Procedure

The following parameters were evaluated: age, sex, body mass index (BMI), body fat percentage, excessive BMI loss in percentage (EBMIL), TSH, free T4 (FT4), GH, insulin-like growth factor I (IGF-I), C peptide, and the type of bariatric surgery performed (Roux-en-Y gastric bypass (RYGB) or sleeve gastrectomy (SG)).The patients anthropometric and analytical parameters were evaluated before and 1, 3, 6, 12, 18, 24, 30, 36, and 42 months after surgery. All blood samples were immediately centrifuged, separated, and frozen at −80 °C. Mid-waist circumference was measured as the midpoint between the iliac crest and the lowest rib, with the patient in the upright position. Total body fat was calculated through bioelectrical impedance analysis (BIA). The primary endpoint was circulating TSH.

### 2.3. Assays and Other Methods

Serum samples were collected after an overnight fast in the morning between 8:00 a.m. and 9:00 a.m. and stored at −80 °C. Serum TSH (mIU/L) was measured by a two-site chemiluminescent immunoassay (ADVIA Centaur, Siemens, Deerfield, IL, USA) with a sensitivity of 0.01 mIU/L and with intra-assay coefficients of variation of 2.5%, 2.4%, and 2.4% for low, medium, and high serum TSH levels, respectively; and with inter-assay coefficients of variation of 5.3%, 3.4%, and 2.1% for low, medium, and high TSH levels, respectively. Serum FT4 (ng/dL) was measured by a direct chemiluminescent immunoassay (ADVIA Centaur, Siemens) with a sensitivity of 0.1 ng/dL and with intra-assay coefficients of variation of 3.3%, 2.2%, and 2.5% for low, medium, and high plasma FT4 levels, respectively; and with inter-assay coefficients of variation of 2.5%, 4.0%, and 2.3% for low, medium, and high FT4 levels, respectively. Serum GH (μg/L) was measured by a solid-phase, two-site chemiluminescent enzyme immunometric assay (Immulite, EURO/DPC, Llanberis, UK) with a sensitivity of 0.01 μg/L and with intra-assay coefficients of variation of 5.3%, 6.0%, and 6.5% for low, medium, and high plasma GH levels respectively; and with inter-assay coefficients of variation of 6.5%, 5.5%, and 6.6% for low, medium, and high GH levels, respectively. IGF-1 (ng/mL) was determined by a chemiluminescence assay (Nichols Institute, San Clemente, CA, USA) and with intra-assay coefficients of variation of 4.8%, 5.2%, and 4.4% for low, medium, and high IGF-1 levels, respectively; and with inter-assay coefficients of variation of 7.7%, 7.4%, and 4.7% for low, medium, and high plasma IGF-I levels, respectively. Plasma glucose (mg/dL) was measured with an automatic glucose oxidase method (Roche Diagnostics, Mannheim, Germany). All samples from a given subject were analyzed in the same assay run.

### 2.4. Calculations

Excess body mass index loss (EBMIL) was calculated using the formula: ((preoperative BMI − current BMI)/(preoperative BMI − 25)) × 100(1)

### 2.5. Statistical Analysis

Continuous variables are expressed as mean ± standard error (SE) and/or median and interquartile range (IR). Categorical data are expressed as frequency and percentages. For comparisons between control subjects and obese patients, Student’s T-test was used for normally distributed data and the Mann-Whitney test was used for comparison of medians. The Wilcoxon test was used to compare the preoperative and 12 months postoperative values in obese patients. For categorical comparisons the χ^2^ test was used. 

Generalized estimating equations (GEE) models, with the autoregressive correlation structure, were used to evaluate the trajectory of post-surgical TSH, as well as to determine factors associated with changes in TSH values after surgery. In multivariable analyses, basal BMI and post-surgery percentage of excess weight loss, GH, IGF-1, and free T4 values were included as independent variables. To test the hypothesis that TSH trajectories would differ based on pre-surgical BMI, the corresponding interaction effect, as well as the main effect, were examined.

Statistical analyses were performed with SPSS, version 24.0, and R, version 3.5.1, with the package geepack added. *p*-values < 0.05 were considered as statistically significant.

## 3. Results

### 3.1. Preoperative Characteristics of the Study Population and the Control Group

Among the 129 patients studied, 92 were women and the mean age was 46.6 ± 0.8 years (Table 1). The patient group presented a mean preoperative BMI of 49.3 ± 0.7 kg/m^2^ (Table 1). Fifty-one percent of the patients had diabetes before surgery; a percentage that decreased to 7.75% after surgery. The surgical procedures performed were RYGB (68.2% of patients) and SG (31.8% of patients). Among the 31 controls studied, 21 were women and the mean age was 44.4 ± 1.7 years (Table 1). The control group presented a mean BMI of 24.1 ± 0.7 kg/m^2^. The two groups had similar sex and age as designed by the matching criteria. The age and adiposity indices of the controls and obese patients are shown in Table 1. 

### 3.2. Fasting Serum Levels

Hormones, fasting glucose, lipids, and C-reactive protein results (mean ± SE; median, interquartile ranges) are shown in Table 2. Fasting TSH levels were higher in the obese group than in healthy controls; 3.3 ± 0.2 vs. 2.1 ± 0.2 for the obese and control group, respectively. Fasting FT4 levels were higher in the obese group than in the healthy controls; 1.47 ± 0.02 vs. 1.1 ± 0.01 for the obese and control group, respectively. Fasting glucose levels were higher in the obese group than in the healthy controls; 105.7 ± 2.6 vs. 89.5 ± 1.4 for the obese and control group, respectively. Fasting IGF-I levels were lower in the obese group than in the healthy controls; 90.0 ± 4.2 vs. 139.1 ± 7.9 for the obese and control group, respectively.

### 3.3. Evolution over Time of the Clinical and Analytical Parameters

The anthropometric, hormonal and biochemical parameters in obese patient before and 12 months after BS are presented in Table 3. The mean EBMIL in percentage 12 months after BS was 72.7 ± 2.1% and the mean BMI decrease after surgery was 17.2 ± 0.6 kg/m^2^. Fasting TSH levels significantly decreased in the obese patients after surgery induced weight loss; 3.3 ± 0.2 vs. 2.1 ± 0.2 for the obese patients before and 12 months after surgery, respectively. 

Figure 1 shows the TSH values (median (IR)) in control subjects and obese patient before and 12 months after surgery. Fasting TSH levels were higher in the obese group than in healthy controls. Fasting TSH levels significantly decreased in the obese patients 12 months after surgery-induced weight loss.

Figure 2 shows the evolution over time of fasting TSH (median (IR)) before and after surgery (0, 1, 3, 6, 12, 18, 24, 30, 36, 42 months) in obese patients. The results show a progressive decrease in TSH. 

In Table 4, the results of sequential generalized estimating equation (GEE) models that examined the trajectory of TSH values after surgery are summarized. A significant trend for decrease in TSH values was determined (*p* < 0.001), estimating a mean decrease around 0.034 units per month of follow-up. Preoperative BMI values were not significantly associated with mean TSH values at follow-up, nor with the rate of decrease over time. The percentage of excess weight loss at each visit was the only variable significantly associated with mean TSH values, with a higher excess weight loss associated with lower TSH values (*p* < 0.001). None of the GH, IGF-1 and FT4 values at each visit were found to be significantly associated with TSH determinations. 

## 4. Discussion

The main results of this study are that circulating TSH levels were found to be increased in patients with morbid obesity without a history of thyroid disease and that weight loss after BS in patients with morbid obesity without a history of thyroid disease promotes a decrease of TSH. This decrement of TSH is progressive over time following surgery. The decrease in TSH is significantly associated with excess BMI lost. The present study suggests that the increased TSH found in obesity is secondary to the obese state. Furthermore, from a clinical perspective, our study highlights the difficulty of diagnosing mild thyroid hormone deficiency in obesity.

Thyroid hormone levels have been reported to be normal, increased, and decreased in obese Patients [11,15]. In agreement with our results, most studies have found increased circulating TSH levels in patients with extreme obesity. Rotondi et al. [12] reported increased TSH levels in a group of 350 morbid obese patients when compared with a group of 50 healthy normo-weight subjects. Reinehr et al. [13] have found increased circulating TSH levels in a group of 118 obese children when compared with 107 healthy children of normal weight. The degree of obesitycorrelated with TSH values. Valdes et al. [14] found increased TSH levels in an ample group of patients with morbid obesity, suggesting that reference values for TSH may be inadequate to define hypothyroidism in persons with morbid obesity. The management of hypothyroidism is complex in the BS patient, as the levothyroxine requirements are mainly dependent on lean body mass, and not on fat mass [21]; moreover, levothyroxine dosage could be modified due to malabsorption after BS and the decrease in BMI and lean body mass [22,23].

The alteration of thyroid function and the effect of BS on postoperative thyroid function evolution are still not fully understood. Several studies have found different results regarding the variation of TSH after weight loss surgery and the relationship between TSH variation and weight loss [15,16,17,18,19,20]. Most [16,17,19], but not all [15,20], studies evaluating the evolution of TSH after weight loss surgery have also found a decrease in circulating TSH after the intervention. In agreement with our results, Guan et al. [16] performed a systematic review and meta-analysis of the effect of BS on thyroid function in obese patients and found that BS was associated with a significant decrease in circulating TSH levels. Neves et al. [19] performed a retrospective observational study of 949 euthyroid patients and found that BS promotes a significant decrease in circulating TSH that is significantly greater in patients with high-normal TSH and is independently associated with excess body weight loss after surgery. On the other hand, Dall’Asta et al. [20] evaluated 99 healthy controls and 258 obese subjects before and after weight loss through gastric banding and found that circulating TSH levels remained steady. Zhang et al. [15] performed a retrospective study of 117 patients, followed up for 36 months after Laparoscopic Roux-en-Y Gastric Bypass Surgery, and found that TSH levels remained stable. The differences between studies could be due to the type of BS performed, the characteristics of the control group and the studied patients, or to the different statistical power of the studies.

The mechanism and the clinical implications of TSH elevation in obesity remains unknown. Various pathophysiological mechanisms underlie the relationship of subclinical hypothyroidism to energy homeostasis and adiposity [24]. These include the direct role of TSH in brown adipose tissue and thermogenesis [25]. TSH receptors are less expressed on adipocytes of obese vs. lean individuals [26]. This reduced TSH receptor expression may induce down-regulation of thyroid hormone receptors and thyroid hormone action, thereby further increasing plasma TSH concentration and constituting a condition of peripheral thyroid hormone resistance [26]. The increase in TSH may represent a compensatory activation of hypothalamus–pituitary–thyroid axis in response to excessive body weight [11]. In agreement with these data, in severe obesity, short-term weight loss reveals a positive connection between resting energy expenditure and thyroid hormones [27]. In the present article, increased free thyroxine was found in obese subjects that decreased after weight loss in accordance with the hypothesis that the increase in TSH may represent a compensatory activation of the hypothalamus–pituitary–thyroid axis [11], while similarly, increased free thyroxine in obesity has been found in other studies [13].This activation appears to be mediated, at least in part, by the hypothalamic or pituitary effects of leptin [28]. On the contrary, Marzullo et al. [29] found that obesity increases the susceptibility to harbor autoimmune thyroid disease with an emerging role for leptin as a peripheral determinant, suggesting that obesity is a pathogenic factor for organic thyroid disease. 

TSH values follow a seasonal pattern, decreasing during the summer, and then increasing during the winter. These annual variations in TSH secretion should be considered for the interpretation of results [30,31]. In addition, there is a robust circadian variation in TSH secretion [32,33]. The mechanisms of TSH decrease after BS remain unclear. This decrement in TSH is probably weight loss mediated and is not due to an intrinsic effect of BS. A decrement of TSH levels was found in obese patients after exercise, behavior therapy, and nutrition-induced weight loss [34], and the decrease in circulating TSH has been found to be independently associated with excess body weight loss after surgery [19]. One of the most likely explanations is the decrease in leptin levels following BS and weight loss [35]. With a decreasing body fat, the decreasing leptin circulating levels [35] reduce the central stimulation of the thyroid axis [28] and promote a decrease in TSH. In addition, the reduced TSH receptor expression is reversed by weight loss, improving TSH resistance [26]. In any case, our results show the decrease in circulating TSH to be independently associated with excess body weight loss after surgery, suggesting that the decrement in TSH is mainly weight mediated. There is an important relationship between the GH–IGF-I axis and obesity [7,8,9,36,37]. Moreover, there is a complex and not yet fully understood relationship between the GH–IGF-I axis and the pituitary–thyroid axis [38]. Thyroid hormones act at many sites from the hypothalamic control of GH release to the tissue expression of IGF-I and its binding proteins [39]. GH therapy is associated with an increase in thyroid volume in GH-deficient patients and a tendency to develop thyroid nodules related to circulating IGF-I [40]. GH modulates the circulating thyroid hormones values [41]. A stimulatory effect of IGF-I on the proliferation of pituitary cells in culture has been demonstrated [42]. Interestingly, there is a synergistic TSH/IGF-I receptor cross-talk that activates extracellular kinases in different cell types [43]. Despite exploring the possible correlation between the GH–IGF-I axis and TSH, we were unable to find any important correlation.

From a clinical perspective, our study highlights that whatever the mechanism underlying increased TSH in obesity, it is difficult to identify obese patients who are affected by mild hypothyroidism [10]. It seems reasonable to suggest that thyroid hormone deficiency could be suspected in obese patients with lightly elevated TSH levels only after measuring circulating levels of thyroid hormones and thyroid autoantibodies and after having detected evidence of impaired thyroid hormone activity at a tissue level [10]. Values approaching the upper-limit of the TSH reference range may not represent subclinical hypothyroidism, but a compensatory response to extreme obesity and persons with extreme obesity might be inappropriately classified if the reference ranges of normality of TSH for the normal-weight population are applied to them [14]. Furthermore, our results do not support the decrease in the upper limit of circulating TSH levels, as proposed by several authors [44,45], to the diagnosis of thyroid diseases in morbid obesity.

We must acknowledge some limitations of the present work. First, the relatively small sample size did not allow for the stratification of different subgroups in the analysis. Secondly, the absence of free T3 values hinders the interpretation of our findings. Thirdly, we did not consider some variables that could influence the study, such as the concomitant use of other drugs. Fourthly, we did not find any significant difference between men and women in the TSH data, probably due in part to the relatively small sample size of the men’s group. In addition, the TSH values were not normalized for seasonal variations. Nevertheless, there are several strengths to our study. We included sex- and age-matched controls to decrease the chances of misclassifying individuals due to variability in these factors. We evaluated TSH secretion at different time points after BS, as most studies evaluated the variation of TSH only using two temporal moments (before and after surgery).

## 5. Conclusions

This study shows that in euthyroid patients with morbid obesity, weight loss induced with BS promotes a significant decline of the moderately increased TSH levels. This decrement of TSH is progressive over time following surgery and significantly associated with excess BMI loss. The increased TSH found in obesity is secondary to the obese state, and diagnosing mild thyroid hormone deficiency is complex in morbid obesity.

## Figures and Tables

**Figure 1 nutrients-11-01121-f001:**
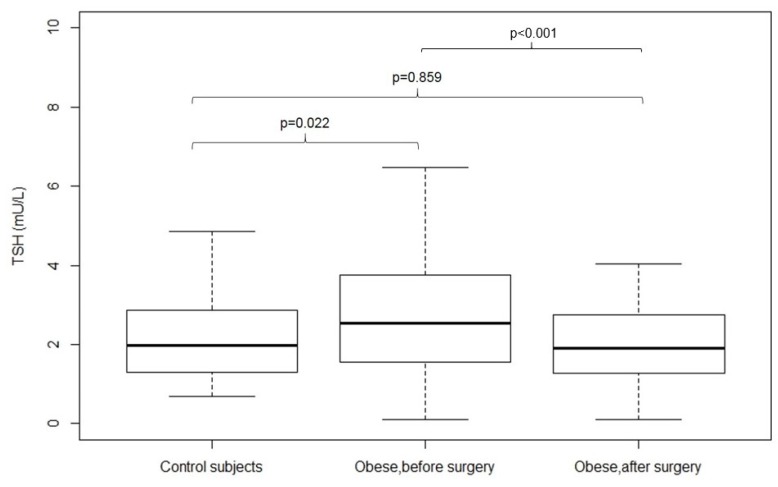
TSH values (median (IR)) in control subjects and obese patient before and 12 months after surgery. Differences were assessed by Student’s *t*-test for normally distributed data and the Mann-Whitney test was used for comparison of medians. *P*-values < 0.05 were considered as statistically significant.

**Figure 2 nutrients-11-01121-f002:**
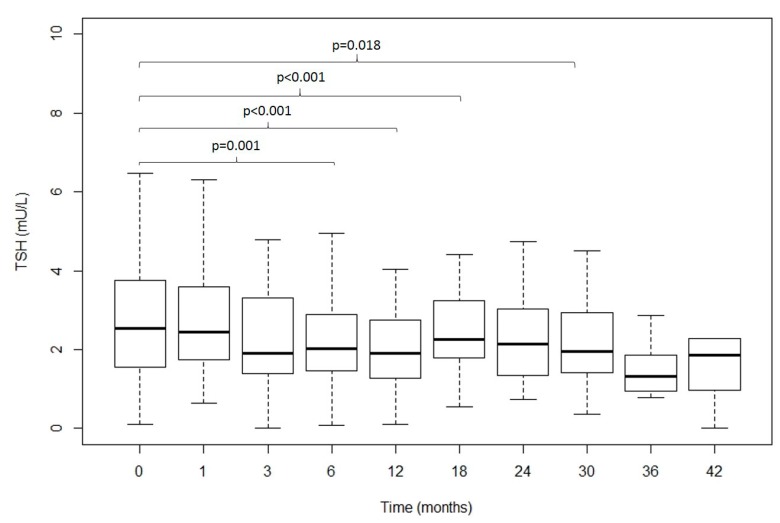
Evolution over time of TSH (median (IR)) before and after surgery (0, 1, 3, 6, 12, 18, 24, 30, 36, 42 months). Differences were assessed by Student’s *t*-test for normally distributed data, and the Mann-Whitney test was used for comparison of medians. *p*-values < 0.05 were considered as statistically significant.

**Table 1 nutrients-11-01121-t001:** Preoperative characteristics of the control subjects and the obese patients (mean ± SE; median, interquartile ranges).

	Control Subjects		Obese Subjects		*p*
	Mean ± SE	Median (IR)	Mean ± SE	Median (IR)	
**Age (years)**	44.4 ± 1.7	42.0 (37.1–53.0)	46.6 ± 0.8	45.8 (43.8–53.7)	0.234
**Sex (*n*, %)**					0.695
Female	21	67.7%	92	71.3%	
Male	10	32.3%	37	28.7%	
**BMI (Kg/m^2^)**	24.1 ± 0.7	23.6 (21.1–25.7)	49.3 ± 0.7	47.8 (43.8–53.7)	<0.001
**Body fat (%)**	26.4 ± 1.5	24.6 (20.6–31.8)	48.8 ± 0.6	50.1 (45.9–52.5)	<0.001
**Diabetes (%)**	0	0%	51	39.5%	<0.001
**HTA (%)**	0	0%	55	42.6%	<0.001
**Type of surgery (%)**					
Roux-en-Y gastric bypass			88	68.2%	
Sleeve gastrectomy			41	31.8%	

BMI: body mass index; HTA: arterial hypertension; differences were assessed by Student’s *t*-test for normally distributed data and the Mann-Whitney test was used for comparison of medians. For categorical comparisons the *χ*^2^ test was used. *P*-values < 0.05 were considered as statistically significant. IR: interquartile range; SE: standard error.

**Table 2 nutrients-11-01121-t002:** Biochemical and hormonal data in control subjects and obese patients, (mean ± SE; median, interquartile ranges).

	Control Subjects		Obese Subjects		*p*
	Mean ± SE	Median (IR)	Mean ± SE	Median (IR)	
TSH (µU/mL)	2.1 ± 0.2	2.0 (1.3–2.9)	3.3 ± 0.2	2.5 (1.5–3.7)	0.022
Free T4 (ng/dL)	1.1 ± 0.01	1.1 (1.1–1.2)	1.47 ± 0.02	1.4 (1.3–1.7)	<0.001
Fasting Glucose (mg/dL)	89.5 ± 1.4	88.0 (85.0–94.0)	105.7 ± 2.6	103.0 (85.0–122.5)	0.002
GH (µg/L)	1.4 ± 0.3	0.5 (0.2–1.9)	1.0 ± 0.2	0.3 (0.1–1.0)	0.256
IGF-I (µg/L)	139.1 ± 7.9	131.0 (100.0–174.0)	90.0 ± 4.2	83.0 (58.8–109.0)	<0.001
Cortisol (µg/dL)	16.5 ± 1.1	15.6 (12.8–19.3)	20.1 ± 2.1	12.8 (8.7–20.45)	0.1
C-Reactive Protein (mg/dL)	0.2 ± 0.1	0.05 (0.02–0.19)	0.9 ± 0.1	0.7 (0.3–1.2)	<0.001

TSH: thyroid-stimulating hormone; T4: Thyroxine; GH: growth hormone; IGF-I: insulin-like growth factor I; differences were assessed by Student’s *t*-test for normally distributed data and the Mann-Whitney test was used for comparison of medians. *P*-values < 0.05 were considered as statistically significant.

**Table 3 nutrients-11-01121-t003:** Anthropometric, biochemical, and hormonal data in obese patients (mean ± SE; median, interquartile ranges).

	Obese Patientsbefore Surgery		Obese Subjects 12 Monthsafter Surgery		Change		*p*
	Mean ± SE	Median (IR)	Mean ± SE	Median (IR)	Mean ± SE	Median (IR)	
BMI (Kg/m^2^)	49.3 ± 0.7	47.8 (43.8–53.7)	32.5 ± 0.7	31.1 (27.8–35.7)	17.2 ± 0.6	16.2 (13.7; 20.5)	<0.001
Weight (Kg)	134.5 ± 2.2	126.0 (115.5–153.4)	88.1 ± 2.1	87.7 (73.7–98.6)	46.7 ± 1.7	42.6 (36.9; 57.0)	<0.001
Body fat (%)	48.8 ± 0.6	50.1 (45.9–52.5)	31.9 ± 1.3	32.8 (23.9–39.3)	18.2 ± 1.2	17.0 (11.7; 20.6)	<0.001
EBMIL (%)			72.7 ± 2.1	73.5 (58.5–83.7)			
TSH (µU/mL)	3.3 ± 0.2	2.5 (1.5–3.7)	2.1 ± 0.2	1.9 (1.2–2.8)	1.2 ± 0.3	0.8 (−0.1; 1.7)	<0.001
Free T4 (ng/dL)	1.47 ± 0.02	1.4 (1.3–1.7)	1.12 ± 0.02	1.1 (1.0–1.2)	0.3 ± 0.0	0.3 (0.2; 0.6)	<0.001
Fasting Glucose (mg/dL)	105.7 ± 2.6	103.0 (85.0–122.5)	90.2 ± 2.1	84.0 (77.0–93.0)	16.7 ± 2.8	16 (0.0; 34.0)	<0.001
GH (µg/L)	1.0 ± 0.2	0.3 (0.1–1.0)	3.4 ± 0.8	1.6 (0.3–5.0)	−2.1 ± 0.9	−0.3 (−4.1; 0.5)	0.014
IGF-I (µg/L)	90.0 ± 4.2	83.0 (58.8–109.0)	115.2 ± 5.0	113.0 (90.2–125.0)	−28.5 ± 6.9	−33.9 (−59.5; −8.0)	<0.001
Cortisol (µg/dL)	20.1 ± 2.1	12.8 (8.7–20.45)	15.4 ± 1.1	15.2 (11.9–17.8)	6.3 ± 4.9	−2.6 (−9.6; 8.1)	0.894
C-Reactive Protein (mg/dL)	0.9 ± 0.1	0.7 (0.3–1.2)	0.2 ± 0.1	0.05 (0.01–0.13)	0.4 ± 1.2	0.6 (0.1; 0.8)	0.001

BMI: body mass index; EBMIL: excessive BMI loss; T4: Thyroxine; IGF-I: insulin like growth factor I; differences were assessed by Wilcoxon test. For categorical comparisons the *χ*^2^ test was used. *P*-values < 0.05 were considered as statistically significant.

**Table 4 nutrients-11-01121-t004:** Generalized estimating equation models examining trajectory of TSH values.

	Model 1		Model 2		Model 3		Model 4		Model 5		Model 6		Model 7	
	B	SE	B	SE	B	SE	B	SE	B	SE	B	SE	B	SE
**Main Effects**														
Linear time (months after surgery)	−0.034	0.009 ***	−0.033	0.009 ***	0.067	0.084	0.001	0.014	−0.021	0.036	−0.019	0.036	−0.019	0.038
**Preoperative values**														
BMI			0.035	0.019	0.050	0.027	0.031	0.019	0.044	0.025	0.038	0.026	0.039	0.026
**Post-surgery values**														
EBMIL							−0.013	0.003 ***	−0.012	0.005 *	−0.012	0.005 *	−0.015	0.007 *
GH									−0.008	0.040	0.001	0.041	0.001	0.043
IGF-I											−0.004	0.003	−0.004	0.003
Free Thyroxine													−0.468	0.595
**Interaction Effects**														
Time × BMI					−0.002	0.002								

B: unstandardized beta; SE: standard error; BMI: body mass index; EBMIL: excessive BMI loss; GH: Growth Hormone; IGF-I: insulin-like growth factor I; * *p* < 0.05; *** *p* < 0.001.

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
