# Peer review of "Effect of Weight Loss after Bariatric Surgery on Thyroid-Stimulating Hormone Levels in Euthyroid Patients with Morbid Obesity"

_nutrients, 2019, doi:10.3390/nu11051121_

Reviewer 1 Report

Comments to the Author

 In this manuscript, the authors investigate the effect of weight loss after bariatric surgery on thyroid-stimulating hormone (TSH) levels in patients with morbid obesity in 129 patients a retrospective observational study. This study demonstrates that TSH decreased over time, and it was associated with EBML in euthyroid subjects with morbid obesity weight loss induced with bariatric surgery promotes a significant decrease in the TSH compared to preoperative levels. The follow-up TSH measurement until 42 months makes this study exciting and novel. The results are of interest, and the manuscript is overall well-written. However, there are some suggestions and concerns which should be addressed.

 Major comments

1. The tile may include ‘euthyroid’; since these subjects fall under this condition. 

2. Abstract: include FreeT4 also higher in morbid obese condition; is that also decline after bariatric surgery? If so mention correctly in the abstract.  3. Patient and methods: It should be included 50% of the obese subjects are diabetic (as per table 1 in the results section). It is interesting to know and include in manuscript; post-bariatric surgery does those diabetic subjects changed their medication regime; if data available it may be included in the manuscript.

4. This study has higher % of Female subjects compared to men, did authors find any gender difference at least in TSH decline after bariatric surgery. Since the influence of sex steroids have been shown in literature in the contest of TSH axis. It is interesting to include in the manuscript if authors found any gender difference in the data; if there is no such difference also mention in limitations paragraph.

5. The levels of TSH follow a circadian pattern and seasonal pattern in mammals; did all the time point sample collected at the same time of the day at different months? Make sure in method sections to include the time of collections. In continuation, figure 1 data follows a seasonal pattern; even it is not the aim of the study. It is interesting to include some studies on the circadian and seasonal pattern of TSH either in introduction or discussion and includes in limitation these data not normalized for seasonal variations.

6. IGF-1 levels significantly increased (Table 3), is there any cross-talk between TSH and IGF-1; shed some literature on those aspects in the discussion.

  Minor Comments

1. Abstract Line50-51” Diagnosing hypothyroidism is complex in obesity” seems that sentence incomplete.

2. Abstract EBMIL must be abbreviated at first instance.

3. Figure legends must be improved, and tables must have footnotes about the parameters discussed, and the statistics applied.

4. There are many other minor errors of syntax and grammar throughout the text, which need to be fixed.

5. I found few supplementary figures are repetitive and that significance even not mentioned in the main manuscript anywhere; for what those included? Mention on the appropriate place in the main manuscript where readers need to check supplementary files for further understanding.

Author Response

REPLY TO REVIEWERS

 The changes and additional text are highlighted in red.

 Reviewer 1:

 Thank you for your comments. We appreciate the referee’s positive comments regarding our work.

  Major comments

 1. Following the reviewer’s recommendations, we have included “euthyroid” in the title. Thank you for your comment.

Page 1, line 4.

 2. Following the reviewer’s recommendations, we have included in the abstract the FreT4 values as you suggest. FreT4 levels significantly decreased in the obese patients after surgery; 1.47±0.02 vs1.12±0.02 before and 12 months after surgery, respectively

Page 2, line 53.

 3. Following the reviewer’s recommendations, we have included these data in the article. Prior to surgery, 51% of the obese patients had diabetes, after surgery the percentage decreased to 7.75%. Unfortunately, we do not have the exact data about the medication regimen of patients. Thank you for your comment.

Page 4, line 122 and page 5, line 198.

 4. We appreciate the reviewer’s comment regarding the influence of sex steroids on TSH. As you correctly point out, the reference interval for TSH varies significantly by sex, with women having significantly higher TSH values than men [1]. We did not find any significant difference between men and women in the TSH data, probably due in part to the relatively small sample size of the men’s group. We have included a comment in the limitations paragraph, as you suggest.

Page 12, line 387.

 5. Following the reviewer’s recommendations, we have included in the methods section the time of collection of the samples. All samples were collected after an overnight fast, in the morning between 8.00 and 9.00 a.m. We have included some studies on the circadian and seasonal pattern of TSH in the discussion. TSH values follow a seasonal pattern, decrease during the summer, while they increase during the winter. These annual variations in TSH secretion should be borne in mind for the interpretation of results[2] [3]. In addition, there is a robust circadian variation on TSH secretion [1] [4]. We have included in the limitations that TSH values were not normalized for seasonal variations.

Page 4, line 144. Page 11, line 347. Page 12, line 389

 6. Following the reviewer’s recommendations we have included in the discussion some data about the cross-talk between the TSH-T4 axis and the GH-IGF-I axis. Thank you for your comment. There is a complex and not yet fully understood relationship between the hypothalamic-GH-IGF-I axis and hypothalamic-pituitary-thyroid axis [5]. Thyroid hormones act at many sites from the hypothalamic control of GH release to the tissue expression of IGF-I and its binding proteins (IGFBPs)[6]. GH therapy is associated with an increase in thyroid volume in GH deficiency patients [7].There is a tendency to develop thyroid nodules, mainly related to pre-therapy IGF-I circulating values [7]. A stimulatory effect of IGF-I on proliferation of mouse pituitary cells in serum-free culture has been demonstrated [8]. Interestingly, there is a synergistic TSH/IGF-I receptor cross-talk that activates extracellular kinases in multiple cell types [9].

Page 12, line 360.

 Minor comments

 1.  Following the reviewer’s recommendations, we have deleted the unclear sentence.  Thank you for your comment.

Page 2, line 58.

 2. Following the reviewer’s recommendations, we have included EBMIL at first mention

Page 2, line 51.

 3.Following the reviewer’s recommendations, we have improved the figure legends and tables, including footnotes about the parameters discussed, and the statistics applied.

Pages 6, 7, 8, 9, 10.

 4. Following the reviewer’s recommendations, we have revised the mistakes in the text and improved the English.

Pages 2-13.

 5. Following the reviewer’s recommendations, we have avoided being repetitive with the supplementary figures and we have deleted the supplementary material. Thank you for your comment.

   References

 1.         Ehrenkranz, J.; Bach, P.R.; Snow, G.L.; Schneider, A.; Lee, J.L.; Ilstrup, S.; Bennett, S.T.; Benvenga, S. Circadian and Circannual Rhythms in Thyroid Hormones: Determining the TSH and Free T4 Reference Intervals Based Upon Time of Day, Age, and Sex. Thyroid 2015, 25, 954-961, doi:10.1089/thy.2014.0589.

2.         Yoshihara, A.; Noh, J.Y.; Watanabe, N.; Iwaku, K.; Kunii, Y.; Ohye, H.; Suzuki, M.; Matsumoto, M.; Suzuki, N.; Sugino, K., et al. Seasonal Changes in Serum Thyrotropin Concentrations Observed from Big Data Obtained During Six Consecutive Years from 2010 to 2015 at a Single Hospital in Japan. Thyroid 2018, 28, 429-436, doi:10.1089/thy.2017.0600.

3.         Das, G.; Taylor, P.N.; Javaid, H.; Tennant, B.P.; Geen, J.; Aldridge, A.; Okosieme, O. Seasonal Variation of Vitamin D and Serum Thyrotropin Levels and Its Relationship in a Euthyroid Caucasian Population. Endocr Pract 2018, 24, 53-59, doi:10.4158/EP-2017-0058.

4.         Roelfsema, F.; Pijl, H.; Kok, P.; Endert, E.; Fliers, E.; Biermasz, N.R.; Pereira, A.M.; Veldhuis, J.D. Thyrotropin Secretion in Healthy Subjects Is Robust and Independent of Age and Gender, and Only Weakly Dependent on Body Mass Index. The Journal of Clinical Endocrinology & Metabolism 2014, 99, 570-578, doi:doi:10.1210/jc.2013-2858.

5.         Giavoli, C.; Profka, E.; Rodari, G.; Lania, A.; Beck-Peccoz, P. Focus on GH deficiency and thyroid function. Best Pract Res Clin Endocrinol Metab 2017, 31, 71-78, doi:S1521-690X(17)30003-9 [pii]

10.1016/j.beem.2017.02.003.

6.         Rodriguez-Arnao, J.; Miell, J.P.; Ross, R.J. Influence of thyroid hormones on the GH-IGF-I axis. Trends Endocrinol Metab 1993, 4, 169-173, doi:1043-2760(93)90107-P [pii].

7.         Curto, L.; Giovinazzo, S.; Alibrandi, A.; Campenni, A.; Trimarchi, F.; Cannavo, S.; Ruggeri, R.M. Effects of GH replacement therapy on thyroid volume and nodule development in GH deficient adults: a retrospective cohort study. Eur J Endocrinol 2015, 172, 543-552, doi:EJE-14-0966 [pii]

10.1530/EJE-14-0966.

8.         Oomizu, S.; Takeuchi, S.; Takahashi, S. Stimulatory effect of insulin-like growth factor I on proliferation of mouse pituitary cells in serum-free culture. J Endocrinol 1998, 157, 53-62.

9.         Krieger, C.C.; Perry, J.D.; Morgan, S.J.; Kahaly, G.J.; Gershengorn, M.C. TSH/IGF-1 Receptor Cross-Talk Rapidly Activates Extracellular Signal-Regulated Kinases in Multiple Cell Types. Endocrinology 2017, 158, 3676-3683, doi:4080141 [pii]

10.1210/en.2017-00528.

 Reviewer 2 Report

The authors have retrospectively assessed changes in thyroid function of patients with morbid obesity who had undergone bariatric surgery procedures. They found a reduction in TSH levels that correlated with the excess BMI loss but not with BMI. 

 ·      Minor comments

1.     Lines 50-51. “Diagnosing hypothyroidism is complex in obesity.” Please clarify. Did the authors mean that standard reference values for TSH might be not appropriate for obese patients? 

2.     Line 82. “All of the studies”. Which studies?

3.     Line 85. Please clarify the number of patients included in this study. Were 47 patients (i.e. 176-129) and 145 controls (i.e. 176-31) excluded? Why?

4.     Table 1. Please define the abbreviation HTA. 

5.     Table 1. The comparison between patients and controls regarding surgical procedures has no sense, since only patients underwent them. 

6.     Table 2. Obese patients had lower total cholesterol and LDL levels compared to controls. Please clarify. 

7.     Table 3. Fasting insulin levels did not change after bariatric surgery as expected. Please comment. 

8.     Figures 1 and 2. I suggest to show significant P values for comparisons with control subjects (Figure 1) or the baseline (Figure 2). 

9.     Patients and controls. Was thyroid autoimmunity biochemically and ultrasonographically evaluated?

10.  Results. As bariatric surgery can reverse diabetes mellitus, it would be interesting to know what was the post-operative prevalence of diabetes. 

11.  I suggest a careful revision of some sentences (for instance, lines 48 and 286: “induced by”; lines 275-276: “for diagnosing hypothyroidism in patients with morbid obesity”; line 284: “two time points”),  and of punctuation (for instance, at line 247). 

12.  One limitation of this study is the lack of body composition assessment. In this regard an Italian study showed that levothyroxine requirements in hypothyroid patients are dependent on lean body mass, not on fat mass [J Clin Endocrinol Metab. 2005 Jan;90(1):124-7]. 

13.  Also, it important to say that bariatric surgery may results in increase of levothyroxine requirements in hypothyroid patients [Obes Surg. 2017 Jan;27(1):78-82]. These two papers are worth of being cited.

 Author Response

REPLY TO REVIEWERS

 The changes and additional text are highlighted in red.

 Reviewer 2

 Thank you for your comments. We appreciate the referee’s positive comments regarding our work.

  Minor comments

 1. Following the reviewer’s recommendations, we have deleted this unclear sentence. Thank you for your comment.

Page 2, line 58.

 2. Following the reviewer’s recommendations, we have clarified this sentence

Page 3, line 118.

 3. Following the reviewer’s recommendations, we have clarified the number of patients included in the study. Thank you for your help to avoid this mistake.

Page 3, line 120.

 4. Following the reviewer’s recommendations, we have defined the abbreviation HTA (arterial hypertension).

Page 6, line 209.

 5. Following the reviewer’s recommendations, we have deleted this comparison.

Page 6, table 1.

 6. Thank you for your interesting comment. We have not included aspects of treatment in the study, and in fact 33.33% of the obese patients were undergoing treatment for elevated lipid levels, and none of the controls. Following your suggestion, we have erased these data from the table, because as you correctly point out they could be confusing.

Page 6, tables 2 and 3.

 7. Thank you for your interesting comment. As previously mentioned, we have not included aspects of medical treatment in the present study. Our obese group included a substantial number of diabetic patients, even those treated with insulin. Following your suggestion, we have erased these data from the table, because as you correctly point out, they could be confusing.

Page 6, tables 2 and 3.

 8. Following the reviewer’s recommendations, we have shown the significant P values in figures 1 and 2.

Pages 8 and 9.

 9. Thank you for your interesting comment. Thyroid autoimmunity was not evaluated systematically either biochemically or ultrasonographically.

 10. Following the reviewer’s recommendations, we have included the post-operative prevalence of diabetes.

Page 5, line 198.

 11. Following the reviewer’s recommendations, we have made a careful revision of some sentences and punctuation.

Pages 2-13.

 12. Following the reviewer’s recommendations, we have included this paper in the references of the article and made a brief comment.

Santini, F., A. Pinchera, et al. (2005). "Lean body mass is a major determinant of levothyroxine dosage in the treatment of thyroid diseases." J Clin Endocrinol Metab 90(1): 124-127.[1]

Page, 12 line .

 13. Following the reviewer’s recommendations, we have included this paper and a similar one in the references of the article, and made a brief comment.

 Fallahi, P., S. M. Ferrari, et al. (2017). "TSH Normalization in Bariatric Surgery Patients After the Switch from L-Thyroxine in Tablet to an Oral Liquid Formulation." Obes Surg 27(1): 78-82.[2]

Pedro, J., F. Cunha, et al. (2018). "The Effect of the Bariatric Surgery Type on the Levothyroxine Dose of Morbidly Obese Hypothyroid Patients." Obes Surg  28(11): 3538-3543.[3]

  References

 1.         Santini, F.; Pinchera, A.; Marsili, A.; Ceccarini, G.; Castagna, M.G.; Valeriano, R.; Giannetti, M.; Taddei, D.; Centoni, R.; Scartabelli, G., et al. Lean body mass is a major determinant of levothyroxine dosage in the treatment of thyroid diseases. J Clin Endocrinol Metab 2005, 90, 124-127, doi:jc.2004-1306 [pii]

10.1210/jc.2004-1306.

2.         Fallahi, P.; Ferrari, S.M.; Camastra, S.; Politti, U.; Ruffilli, I.; Vita, R.; Navarra, G.; Benvenga, S.; Antonelli, A. TSH Normalization in Bariatric Surgery Patients After the Switch from L-Thyroxine in Tablet to an Oral Liquid Formulation. Obes Surg 2017, 27, 78-82, doi:10.1007/s11695-016-2247-4

10.1007/s11695-016-2247-4 [pii].

3.         Pedro, J.; Cunha, F.; Souteiro, P.; Neves, J.S.; Guerreiro, V.; Magalhaes, D.; Bettencourt-Silva, R.; Oliveira, S.C.; Costa, M.M.; Queiros, J., et al. The Effect of the Bariatric Surgery Type on the Levothyroxine Dose of Morbidly Obese Hypothyroid Patients. Obes Surg 2018, 28, 3538-3543, doi:10.1007/s11695-018-3388-4

10.1007/s11695-018-3388-4 [pii].